# Influence of Cross-Section and Pitch on the Mechanical Response of NiTi Endodontic Files under Bending and Torsional Conditions—A Finite Element Analysis

**DOI:** 10.3390/jcm11092642

**Published:** 2022-05-08

**Authors:** Victor Roda-Casanova, Antonio Pérez-González, Alvaro Zubizarreta-Macho, Vicente Faus-Matoses

**Affiliations:** 1Department of Mechanical Engineering and Construction, Universitat Jaume I, 12071 Castelló de la Plana, Spain; vroda@uji.es (V.R.-C.); aperez@uji.es (A.P.-G.); 2Department of Dentistry, Alfonso X el Sabio University, 28691 Madrid, Spain; 3Department of Orthodontics, University of Salamanca, 37008 Salamanca, Spain; 4Department of Stomatology, Faculty of Medicine and Dentistry, University of Valencia, 46010 Valencia, Spain; vfaus@clinicafaus.com

**Keywords:** endodontic file, cross-section, pitch, flexural bending, torsion, stress distribution, finite element analysis

## Abstract

In this article, the effects of cross-section and pitch on the mechanical response of NiTi endodontic files is studied by means of finite element analyses. The study was conducted over a set of eight endodontic rotary files, whose geometry was obtained from combinations of two cross-sections (square and triangular) and four pitches. Each file was subjected to bending and torsional analyses, simulating the testing conditions indicated in the ISO 3630 Standard, in order to assess their stiffness and mechanical strength. The results indicate that endodontic files with a square cross-section have double the stiffness of those with triangular cross-sections, both in terms of bending and torsion. For both loading modes, endodontic files with a triangular cross-section can undergo larger deformations before overload failure than those with a square cross-section: up to 20% more in bending and 40% in torsion. Moreover, under equivalent boundary conditions, endodontic files with triangular cross-sections present a higher fatigue life than those with square cross-sections: up to more than 300% higher for small pitches. The effect of pitch on the stiffness and strength of the file is smaller than that of the cross-section shape, but smaller pitches could be beneficial when using a triangular cross-section, as they increase the bending flexibility, fatigue life, and torsion stiffness. These results suggest a clinical recommendation for the use of files with a triangular-shaped cross-section and a small pitch in order to minimize ledging and maximize fatigue life. Finally, in this study, we reveal the sensitivity of the orientation of files with respect to the bending direction, which must be taken into account when designing, reporting, and interpreting test results under such loading conditions.

## 1. Introduction

The introduction of nickel–titanium alloy (NiTi) for the manufacturing of root canal instruments entailed a great revolution in the field of endodontics, as the consequent endodontic files decreased the incidence of iatrogenic complications [1,2]. However, despite the continuous mechanical and chemical improvements made by manufacturers, the failure of endodontic files during root canal treatments remains a concern for clinicians [3], as the incidence of their fracture still ranges from 0.09% to 5% [4,5].

The fracture of rotary instruments occurs mainly due to two different mechanisms, usually referred to as torsion overload and flexural fatigue [6,7]. On one hand, the torsion overload failure mechanism corresponds to a static failure that typically occurs when the tip of the endodontic file becomes blocked in the root canal whilst the instrument continues rotating [8]. In static failure, the file fails because the stress value reaches the elastic limit of the material, such that the file undergoes permanent deformation and finally fractures. On the other hand, flexural fatigue is a failure mechanism produced mainly by the alternating compressive and tensile stresses and strains that appear in any point of a file rotating inside a curved root canal [8,9]. This type of fatigue failure results in a sudden fracture of the file after a certain number of rotations, even if the stress levels are far below the elastic limit of the material, due to the nucleation and progression of small cracks in some stressed sections of the file. Thus, bending and torsion are essential conditions to evaluate the mechanical behavior of endodontic instruments [10]. The unexpected failure of NiTi endodontic files may condition the outcome of the root canal treatment by blocking the advancement of disinfecting agents beyond the fractured instrument [11,12,13], which may lead to subsequent pulp necrosis and the formation of periapical lesions [14], or decrease the success rate of root canal treatment of teeth with periapical pathology [15]. In addition, extraction of the fractured NiTi endodontic rotary file from the root canal system requires root dentin removal to provide access to the fractured instruments [16]. This causes a loss of dentin tissue, which can negatively affect the structural integrity of the tooth [17]. Furthermore, it can lead to root perforation and increase the risk of vertical root fracture, especially in the apical third [16]. For these reasons, a better understanding of the independent and combined effects of the different parameters that affect these failure mechanisms is desirable, and additional research must be addressed to this end.

Several works have been conducted to analyze the influence of both the NiTi alloy [18] and the geometrical parameters on the torsional and bending resistance of endodontic instruments. Both the chemical composition and crystalline structure of the NiTi alloy have been studied, and it has been shown that they highly influence the strength of the endodontic file [19]. In particular, endodontic rotary systems with a higher concentration of the martensitic phase and manufactured using electropolishing, ion implantation, cryogenic treatment, and heat treatments improve the mechanical behavior of NiTi endodontic rotary files, increasing their cyclic fatigue resistance. The geometric parameters of the endodontic files have also been reported to influence the instrument’s performance, including the taper and apical diameter [20], cross-section design [21,22], flute length, helix angle, and pitch [23]. The influence of these variables has been analyzed using static and dynamic custom-made cyclic fatigue testing devices, which have not been submitted to a standardization normative, and do not allow for independently assessing the influence of each geometric parameter associated with flexural fatigue or torsional overload. There are other standardized testing devices, such as those described in ISO 3630-1:2008 [24], which allow for the independent assessment of both torsional and bending phenomena, although their capability to reproduce the actual operating conditions of endodontic files has not yet been verified.

Computer simulation has proven to be an interesting tool for studying the failure of endodontic rotary files. In the simplest cases, analytical methods can be used for such a purpose, which are usually based on the small strain theory of elasticity. In this line, Zhang et al. [25] have analyzed the mechanical behavior of NiTi endodontic files under torsional and bending loads. Tsao et al. [26] have developed analytical models to study the flexibility of NiTi instruments subjected to bending loads. These analytical models have the advantage of being fast and easy to implement, but their capabilities to consider non-linear behaviors (i.e., material non-linearity) or complex loading scenarios are limited. These limitations can be overcome by using numerical methods such as the finite element method.

The ability of the finite element method to reproduce the results obtained from experimental tests using endodontic rotary files has been proven in several works [7,10,27,28,29], whose main conclusions have been summarized in a recent bibliographical review [30]. This review concluded that the finite element method is a reliable tool for evaluating the behavior of NiTi rotary instruments, and has the advantage of reducing instrument development time and costs. Another important advantage of the finite element method is that it also allows us to assess aspects of the mechanical behavior of the instruments, such as the stress distribution, which are difficult to obtain in laboratory tests [10]. The finite element method has been previously used to analyze the influence of cross-section design and pitch on the stiffness and stress distribution under bending and torsional conditions [10,31,32,33,34,35,36,37]. Appendix B collects detailed information about these previous studies, including their main conclusions and limitations. Some of these studies have used proprietary file models, such as ProTaper, ProFile, Mtwo, and others, which hampers the independent evaluation of parameters such as cross-section geometry, cross-section area, or pitch [10,32,33,34,35]. Other studies have used theoretical file models to avoid this problem, but with some limitations; for example, in [36], the authors analyzed four different cross-sections and three pitch values under torsion, but did not provide detailed information about the material model for the shape memory alloy (SMA) of the files or about the quality of the finite element mesh. In another study, Versluis et al. [33] analyzed the effects of pitch and cross-section geometry on flexural stiffness and stresses using a representative SMA material model. However, the boundary conditions were specified differently to those in ISO 3630-1:2008 [24] and the bending applied was low, leading to maximum von Mises stresses below the initial stress for transformation from austenite to martensite, and, thus, the effect of the super-elasticity of the files was not analyzed; furthermore, torsion behavior was not included in the study. In [37], the effect of cross-section geometry and pitch on the ‘screw-in’ tendency of the files was analyzed, but a linear material model was used for the file. A more recent study investigated different geometric options for the sides of a triangle-shaped cross-section (straight, convex, and concave), as well as the use of files with combinations of these geometries along the file [31], but the pitch effect was not analyzed.

Some of these finite element models are limited in their accuracy, in terms of representing the correct geometry and boundary conditions of the endodontic files, or use simplified material models that are incapable of representing their actual mechanical response under load. In this study, we address all of these partial limitations of previous studies by undertaking a comprehensive analysis of the effects of pitch and cross-section using an accurate finite element model that allows us to simulate the testing conditions of the ISO3630 Standard to the best extent possible. The method used to obtain the parametric geometrical representation of the endodontic instrument and the corresponding finite element mesh has been proposed in our previous work [38]. The use of an accurate numerical model in these tests can foster improvements in new generations of more resistant and flexible endodontic files, reducing the need for expensive and time-consuming experiments in the early design stages. From a clinical perspective, these improvements are expected to reduce the risk of failure of endodontic instruments, thus preventing clinical complications.

The aim of this study was to analyze and compare the effects of the cross-section and the pitch on the mechanical response (in terms of strength and stiffness) of NiTi endodontic files under bending and torsional conditions, similar to those indicated in the ISO 3630 Standard [24], using the finite element method. The study was conducted using a set of eight different endodontic rotary files whose geometries were obtained from combinations of two cross-sections (triangular and square) and four pitches (1mm, 2mm, 4mm, and 8mm). Under these conditions, the following individual objectives were pursued: (i) to develop a finite element model which reproduces the experimental tests conducted in the ISO 3630 Standard; (ii) to conduct a bending analysis of the selected endodontic rotary files, in order to predict the stiffness and strength of the files under static and cyclic loading conditions; and (iii) to conduct a torsional analysis of the selected endodontic rotary files, in order to predict the stiffness and the strength of the files under static loading conditions.

## 2. Materials and Methods

For this study, different endodontic instruments were analyzed using numerical simulation with finite elements. Figure 1 shows the geometries of the eight endodontic files considered. The different geometries were obtained by varying the cross-section (square and triangular) and the pitch ($p_{z}=\left\{1\;\mathrm{mm},2\;\mathrm{mm},4\;\mathrm{mm},8\;\mathrm{mm}\right\}$) of the files. All of them had a total length of $L_{total}=25\;\mathrm{mm}$, the length of their active part was $L_{a}=16\;\mathrm{mm}$, and their tip and shaft diameters were $d_{a}=0.25\;\mathrm{mm}$ and $d_{sh}=1.20\;\mathrm{mm}$, respectively. The taper of the endodontic files was 6%.

The material for all the files was considered to be NiTi, which exhibits a super-elastic stress–strain curve, as shown in Figure 2. Here, $E_{A}$ and $E_{M}$ represent the Young’s moduli of austenite and martensite, respectively. The beginning and end of the loading phase transformation are denoted by $\sigma_{L}^{S}$ and $\sigma_{L}^{E}$, respectively, whereas the beginning and the end of the unloading transformation phase are denoted by $\sigma_{U}^{S}$ and $\sigma_{U}^{E}$. Finally, $\varepsilon_{L}$ represents the uni-axial transformation strain, and $\sigma_{ME}^{E}$ indicates the end of the martensitic elastic regime.

### 2.1. Devices for Experimental Bending and Torsion Analysis

Endodontic files are usually tested in terms of bending and torsional loads, and the typical standardized procedure for these tests has been described in the ISO 3630 Standard [24], as summarized in Figure 3. For the torsion analysis (Figure 3a), the last 3mm at the tip of the endodontic file are inserted inside a clamping jaw. After checking that the endodontic file is properly fixed and aligned with the axis of rotation, the top of the file is rigidly connected to the torsion device. This torsion device is increasingly rotated at angle $\theta_{z}$, and the torsional moment $M_{z}$ is measured using a torquemeter attached to the clamping jaw. The test ends with the failure of the endodontic file. At this point, the maximum rotated angle $\theta_{z,max}$ and maximum torsional moment $M_{z,max}$ are registered.

In a similar way, in the bending analysis (Figure 3b), the last 3mm at the tip of the endodontic file are inserted inside a clamping jaw. After checking that the endodontic file is properly fixed and aligned with the axis of rotation, the bending device is positioned until it contacts the endodontic file. Then, the bending device is increasingly rotated at angle $\theta_{x}$, and the bending moment $M_{x}$ is measured using a torquemeter attached to the clamping jaw. The test ends with the failure of the endodontic file. At this point, the maximum rotated angle $\theta_{x,max}$ and maximum bending moment $M_{x,max}$ are registered.

### 2.2. Definition of the Finite Element Model for the NiTi Endodontic File

Figure 4 shows an example of the finite element model created for the endodontic file simulation experiments, as described in Section 2.1. Here, only the portion of the endodontic file subjected to stresses and strains was considered in the analysis (i.e., the part of the endodontic file inserted into the clamping jaw was not included in the finite element model). The geometry of the endodontic file was generated and then discretized into quadratic finite element tetrahedrons following the meshing procedure developed in our previous work [38]. Using this procedure, the finite element mesh of a endodontic file was automatically built from its geometrical parameters ($d_{sh}$, $d_{a}$, $L_{a}$, $L_{total}$, and $p_{z}$, as shown in Figure 1) and the average element size.

To select the average element size, a mesh sensitivity study was conducted in our previous work [38] for a finite element model of an endodontic file with similar geometry, element type, boundary, and loading conditions, as described in Figure 4. In this study, the variations in the maximum element energy error and energy norm error with respect to the average element size were observed, and it was concluded that an average element size equal to 0.1mm provided a good compromise between accuracy and computational cost. For these reasons, this average element size was used to perform this study, resulting in a finite element model with 89,295 nodes and 58,749 elements.

The super-elastic behavior of the NiTi alloy used to manufacture the endodontic files was modeled using the material model developed by Auricchio [39]. The material properties that characterize this material model were extracted from [10], and are shown in Table 1.

The surface at the fixed end of the endodontic file was defined as a rigid surface (denoted as rigid surface A in Figure 4). This rigid surface was rigidly connected to reference node A, which was used to introduce the boundary conditions for the finite element model. To simulate the effect of the clamping jaw over the endodontic file, all of the degrees of freedom of reference node A were restricted. At the other side of the file, the top surface was also defined as a rigid surface (denoted as rigid surface B in Figure 4). This rigid surface was rigidly connected to reference node B, which was used to define the loading conditions of the model. Two different loading conditions were considered in the analyses, one for the bending analysis and the other for the torsional analysis:In the bending analysis, an increasing displacement was imposed at reference node B in the negative direction of the *y*-axis, until the maximum von Mises stress along the endodontic file $\sigma_{max}$ reached the end of the martensitic elastic regime. As the results of the bending analyses are sensitive to the orientation of the endodontic file with respect to the bending direction, the analysis was conducted in 24 different angular positions, given by a rotation $\varphi_{z}=\left\{0^{\circ },15^{\circ },30^{\circ },\ldots ,360^{\circ }\right\}$ of the endodontic file with respect to the *z*-axis.In the torsional analysis, an increasing rotation was imposed at reference node B along the positive direction of *z*-axis, until the maximum von Mises stress along the endodontic file $\sigma_{max}$ reached the end of the martensitic elastic regime. Here, the results of the analysis do not depend on the orientation of the file.

The finite element model was solved through transient analysis using the large displacements formulation, which was conducted using the ABAQUS software. Hence, material and geometric non-linearities were considered in the study. In each one of these analyses, the rotation at reference node B ($\theta_{x}$ for bending analysis and $\theta_{z}$ for torsional analysis) and the reaction moment at reference node A ($M_{x}$ for bending analysis and $M_{z}$ for torsional analysis) were registered for each analysis frame. The maximum von Mises stress and the maximum principal strain were also retrieved for each analysis frame, using the method indicated in Section A.1, in order to minimize possible numerical singularities in the model. Finally, the bending fatigue life was estimated following the method described in Section A.2, based on the Coffin–Manson relation, considering the material properties indicated in Table 2.

## 3. Results

### 3.1. Bending Analysis

Figure 5 shows the von Mises stress plot for the bending analysis of two representative endodontic files with pitch $p_{z}=4\;\mathrm{mm}$ and analysis angular position given by $\varphi_{z}=0^{\circ }$, for the analysis frame in which the maximum von Mises stress in the model reaches the end of the loading transformation phase $\left(\sigma_{max}=\sigma_{L}^{E}\right)$. Figure 5a shows the von Misses stress plot over an endodontic file with square cross-section and Figure 5b shows the von Misses stress plot over an endodontic file with triangular cross-section. The figure shows that, under these boundary conditions, the highest stresses were located in the apical third of the file.

Figure 6 shows the relationship between the rotation $\theta_{x}$ and the reaction bending moment $M_{x}$ obtained from the bending analysis of the endodontic files with square (Figure 6a) and triangular (Figure 6b) cross-sections and pitch $p_{z}=4\;\mathrm{mm}$. Here, the abscissa axis shows the rotation of the reference node B along the *x*-axis, while the ordinate axis shows the reaction bending moment at reference node A. The figure also shows the points where the maximum von Mises stress in the finite element model reaches the start of the phase transformation, the end of the phase transformation, and the end of the martensitic elastic regime. The curves in the figure exhibit a significant decrease in the slope for a rotation close to 20${}^{\circ }$, corresponding to a change in the stiffness of the file, as the transformation from austenite to martensite progresses in part of the file. As the bending response of an endodontic file is dependent on its orientation (given by the angle $\varphi_{z}$), different curves were obtained for each cross-section. For clarity, only the lower and upper curves are shown for each case, along with another intermediate representative curve. The figures also show the cross-section orientation at the encastré for each case.

Figure 7 shows the bending overload failure mechanism evaluation, which occurs when the maximum von Mises stress in the endodontic file reaches the end of the martensitic elastic regime $\left(\sigma_{max}=\sigma_{ME}^{E}\right)$. On one hand, Figure 7a shows, for each considered pitch and cross-section, the rotation that needs to be applied at the free end of the endodontic files to reach the end of the martensitic elastic regime in the bending analysis. On the other hand, Figure 7b shows, for each considered pitch and cross-section, the maximum bending moment that can be applied at the free end of the endodontic files before they reach the end of the martensitic elastic regime in the bending analysis. As different angular positions were evaluated for each cross-section and pitch, the results shown are the range between the minimum and maximum obtained values. The bold lines represent the mean value within this range.

Figure 7a shows that the maximum rotation was, on average, quite similar for triangular and square cross-sections when the pitch value was larger than 3mm. For these pitch values, it was nearly independent of the pitch, but with a slight tendency to increase with the pitch when using a square cross-section and to decrease when using a triangular cross-section. For pitches below 3mm, files with triangular cross-sections exhibited larger rotations than files with square cross-sections. From Figure 7b, it can be observed that the moment required to bend the square cross-section to failure was almost twice that for the triangular cross-section. The results shown in Figure 7a,b indicate that square cross-sections are more sensitive to the orientation of the file ($\varphi_{z}$) than triangular cross-sections, as the results exhibited larger variability.

Figure 8 shows the bending stiffness of the endodontic rotary files for the austenite and transformation phases. The stiffness in the austenite phase was calculated as the slope of the bending moment–rotation curve before $\sigma_{L}^{S}$, while that in the transformation phase was calculated as the slope of the bending moment–rotation curve between $\sigma_{L}^{E}$ and $\sigma_{ME}^{E}$. In general, it was observed that the stiffness of the endodontic files with square cross-sections was larger than that of the files with triangular cross-sections, both in the austenite and transformation phases. Moreover, the sensitivity to the orientation of the files with square cross-sections was larger than that of those with triangular cross-sections, especially in the austenite phase. The effect of the pitch on the stiffness was negligible for pitches larger than 3mm. With smaller pitches, a reduction in the stiffness was observed, except for the austenite phase with the square cross-section.

Finally, Figure 9 shows the evaluation of the expected fatigue life of the endodontic files when cyclically subjected to a purely reversed bending, which produced a rotation of $\theta_{x}=20^{\circ }$ at the free end of the file. As explained in Section A.2, the bending fatigue life depends on the maximum principal strain in the file. Figure 9a shows the maximum principal strain predicted by the finite element model as a function of the pitch, for both square and triangular cross-sections. In both cases, the effect of file orientation with respect to the bending moment was significant, and the effect of the pitch was noted especially for pitches smaller than near 3mm, for which a decrease in the strain was observed. For the square cross-section, the increase was almost linear; meanwhile, for the triangular cross-section, this increase approximated a logarithmic function. Figure 9b shows the number of cycles that the endodontic files could bear before bending fatigue failure, calculated from the maximum principal strains using the Coffin–Manson relation. It was observed that endodontic files with triangular cross-sections can withstand a larger number of cycles than those with square cross-sections, especially for small pitches.

### 3.2. Torsional Analysis

Figure 10 shows the von Mises stress plot for the torsional analysis of the endodontic files with square (Figure 10a) and triangular (Figure 10b) cross-sections and pitch $p_{z}=4\;\mathrm{mm}$, for the analysis frames in which the maximum von Mises stress in the model reached the end of the loading transformation phase $\left(\sigma_{max}=\sigma_{L}^{E}\right)$. As in the case of the bending analysis, the highest stresses were located near the apical part of the file.

Figure 11 shows the relationship between the rotation $\theta_{z}$ and the reaction torque $M_{z}$, obtained from the torsional analysis of the endodontic files with square (Figure 11a) and triangular (Figure 11b) cross-sections. Here, the abscissa axis shows the rotation of reference node B along the *z*-axis, while the ordinate axis shows the reaction torsional moment measured at reference node A. The figure also shows the points where the maximum von Mises stress in the finite element model reaches the start of the phase transformation, the end of the phase transformation, and the end of the martensitic elastic regime.

Figure 12a shows, for each considered pitch and cross-section, the maximum rotation that needed to be applied at the free end of the endodontic files so that they reached the end of the martensitic elastic regime in the torsional analysis. The results show that the triangular cross-section was able to bear larger rotations before plastic deformation than the square cross-section. The rotation before failure was nearly independent of the pitch with the square cross-section, whereas it increased with the pitch for the triangular cross-section and pitch values between 1mm and 4mm. Figure 12b shows, for each considered pitch and cross-section, the maximum torque that could be applied at the free end of endodontic files before they reached the end of the martensitic elastic regime in the torsional analysis. It was observed that a square cross-section was able to bear almost double the torsional moment of the triangular cross-section. The strength of the files was independent of the pitch for these loading conditions.

Finally, Figure 13 shows the torsional stiffness of the endodontic rotary files for the austenite and transformation phases. The stiffness in the austenite phase was calculated as the slope of the torque–rotation curve before $\sigma_{L}^{S}$, while the stiffness in the transformation phase was calculated as the slope of the torque–rotation curve between $\sigma_{L}^{E}$ and $\sigma_{ME}^{E}$. In general, it was observed that the stiffness of the endodontic files with a square cross-section was larger than that of those with a triangular cross-section, both in the austenite and transformation phases.

## 4. Discussion

In this study, we applied an accurate non-linear finite element model to better understand the effects of the cross-section and pitch of NiTi endodontic files on their mechanical response under bending and torsion loads, according to the ISO 3630 Standard. Finite element analysis has been shown to be a good tool for this type of analysis, providing information about the stress distribution and circumventing experimental variability limitations [24]. Previous research using simulation with the same or similar objectives was first thoroughly analyzed, and the main conclusions and limitations of these studies are summarized in Appendix B, as a reference for further research. The importance of this research is supported by fact that the failure of endodontic files during root canal treatments remains a serious concern for clinicians.

The results of this study demonstrated that, for equal file diameter and taper, the cross-section shape, either triangular or square, has a greater effect than the pitch on the flexural and torsional stiffness of the file. The use of a square cross-section more than doubled the stiffness, compared to that of the triangular cross-section, as explained by the greater second moment of the area of the cross-section. The effect of pitch on stiffness was only appreciable for pitches lower than 3mm, and was more important for triangular than for square cross-sections. When a NiTi file is bent or twisted, according to the conditions of ISO 3630, the super-elastic behavior of the material appears—which is evident from a significant decrease in the stiffness of the file—as a result of the progression of the transformation from the austenite to martensite phase in the most stressed areas of the file (see Figure 6 and Figure 11). Our results indicate that, for a file with a shaft diameter of 1.2mm and 6% taper, this change in stiffness appears when the rotation of the shank end section, with respect to the tip end section, is approximately $20^{\circ }$ in bending or $30^{\circ }$ in torsion. The stiffness of the file decreases by a factor greater than 2 after this transformation point (Figure 8 and Figure 13). The file pitch has the opposite effect on the stiffness for torsion and bending: decreasing the pitch reduces the flexural stiffness, but increases the torsional stiffness. This effect is common for triangular and square cross-sections in the austenite phase, but it is less clear in the transformation phase, where the stiffness is less affected by pitch. This result is in agreement with those obtained in [33,36] for bending and torsion, respectively. As indicated in [33], pitch reduction could benefit both cutting efficiency, due to the higher torsional stiffness, and better adaptation to the canal shape, due to lower bending stiffness.

The obtained stress distributions (Figure 5 and Figure 10) indicate that, for the boundary conditions imposed by the ISO 3630 Standard, the highest stresses were located near the tip of the file (where it is clamped), both in terms of bending and torsion and for both cross-section shapes. The stresses in the proximal part of the file were negligible when the stress corresponding to the end of the loading transformation phase ($\sigma_{max}=\sigma_{L}^{E}$) was reached in the tip of the file. This can be explained by the smaller section at the tip and, in the case of bending, by the higher bending moment in this area.

Static failure under bending was obtained for comparable rotations—close to $40^{\circ }$ for pitch greater than 3mm and ranging between $40^{\circ }$ and $60^{\circ }$, depending on the pitch—for both triangular and square cross-sections (see Figure 7a). However, the bending moment necessary to reach this bending (and, thus, the reaction in the clamp) was quite different, given the difference in stiffness between the cross-section shapes (Figure 7b). This implies greater reaction forces (close to double) in the root canal with the square cross-section than with the triangular cross-section, for comparable bending deformations. The effect of the pitch on bending strength was only significant for pitches below 3mm, where a progressive reduction in strain was observed when the pitch decreased (Figure 9a). This allows for bending of the file to a greater deformation before failure for small pitches, with a corresponding higher expected fatigue life for the same bending deformation (Figure 9b). This effect was especially observed for the triangular cross-section and, to a lesser extent, for the square cross-section. The analysis carried out to estimate the fatigue life also showed that, for the same pitch, the triangular cross-section had a higher expected life than the square cross-section, in agreement with [36], the difference being remarkable for the smallest pitch analyzed (1mm), for which the expected life may be more than three times longer.

Our results showed that the orientation of the bending moment, with respect to the cross-section, had a significant effect on the results, changing the results by up to $19.1^{\circ }$ and 0.97N·mm for the square cross-section and up to $13.0^{\circ }$ and 0.27N·mm for the triangular cross-section. This should be taken into account when designing, reporting, and interpreting experimental bending tests according to ISO 3630.

On the other hand, for torsion, the triangular cross-section files could be rotated to a higher angle before failure than those with a square cross-section, as can be observed from Figure 12. However, due to the difference in stiffness, this failure was reached for a torque less than half that for the square cross-section. The effect of the pitch was opposite to that observed in bending, with a reduction in the pitch leading to a lower strength, as shown by the lower possible rotation before failure, which was also in agreement with the results in [36].

From a clinical perspective, the results obtained in this study suggest that the use of a triangular-shaped cross-section with small pitch for endodontic files could be better for the safe shaping of curved root canals, as its lower stiffness would produce less reaction forces in the channel, thus reducing the possibility of ledging and canal transportation. At the same time, files with a triangular cross-section and 1 mm pitch could exhibit a fatigue life more than double that of files with higher pitches or with a square cross-section. This is accompanied by a lower rotational stiffness, which could be beneficial for improving cutting efficiency [36]. The use of a smaller pitch can only partially compensate for this lower torsional stiffness of the triangular cross-section.

The results obtained in this simulation study refer to the boundary conditions established for the tests described in ISO 3630; however, it should be noted that the stress distribution within the file in these tests is not always comparable to the clinical situation, as the bending of the file is also constrained by contact with the canal walls, resulting in a different deformation, depending on the root curvature. As shown in [38], in a curved canal, the maximum strain is usually located near the highest curvature of the curved root canal axis and the fatigue life is clearly dependent on the radius of curvature. Under the conditions of ISO 3630, the highest curvature of the deformed file is close to the tip, so the conclusions in this study are especially valid for root canals with the highest curvature located near the apical end.

Finally, this work has certain limitations that deserve to be mentioned. This investigation was conducted through theoretical studies, by means of finite element analyses of endodontic rotary files; as such, no experimental tests were conducted. Regarding the investigated endodontic file geometries, all of them had uniform parameters (pitch and cross-section) throughout their entire length, even though there exist endodontic instruments in which these parameters vary through their active length. Finally, the bending fatigue life of the endodontic instruments was assessed considering a fully reversed fatigue phenomenon corresponding to a continuous rotation motion of the file within the root canal. The study of the bending fatigue under other types of motion (e.g., reciprocating and adaptive motions) is left for future research.

## 5. Conclusions

In this study, we simulated the mechanical response of NiTi rotary endodontic files with different cross-sections and pitches using an accurate finite element model under bending and torsion according to the conditions of the ISO 3630 Standard.

From the results obtained, we can conclude that, with equivalent shaft diameter and taper, endodontic files with a square-shaped cross-section have more than double the stiffness of those with a triangular-shaped cross-section under both bending and torsion. The effect of the pitch on stiffness was less significant, but the use of a pitch lower than 3 mm made the files more flexible for bending and stiffer for torsion when using a triangular cross-section, with beneficial effects seen in clinical use. The phase transformation from austenite to martensite led to a significant decrease in file stiffness both in bending and torsion, which was noticeable in the moment versus deformation curve. When the files were deformed under bending or torsion up to failure, a higher angle of rotation was possible before failure for the triangular section, especially in torsion and, for small pitches, in bending. A higher fatigue life can be expected in clinical use with the triangular-shaped cross-section than for the square cross-section under equivalent file deformations, especially with small pitch values. These results suggest a clinical recommendation for the use of files with triangular-shaped cross-sections and small pitch, in order to minimize ledging and maximize fatigue life.

Under the conditions of the ISO 3630 standard, the orientation of the bending plane with respect to the cross-section of the file had a significant effect on the stiffness and the strength of the file. This effect should be taken into account when designing, reporting, and interpreting similar test results.

Further works on this topic could be focused on studying the mechanical response of endodontic instruments with variable parameters (e.g., in terms of pitch and cross-section) throughout their active length. The bending fatigue life of the endodontic files in cases where the loading conditions do not represent a fully reversed fatigue phenomenon (e.g., adaptive or reciprocating motions) also deserves attention in future investigations.

## Figures and Tables

**Figure 1 jcm-11-02642-f001:**
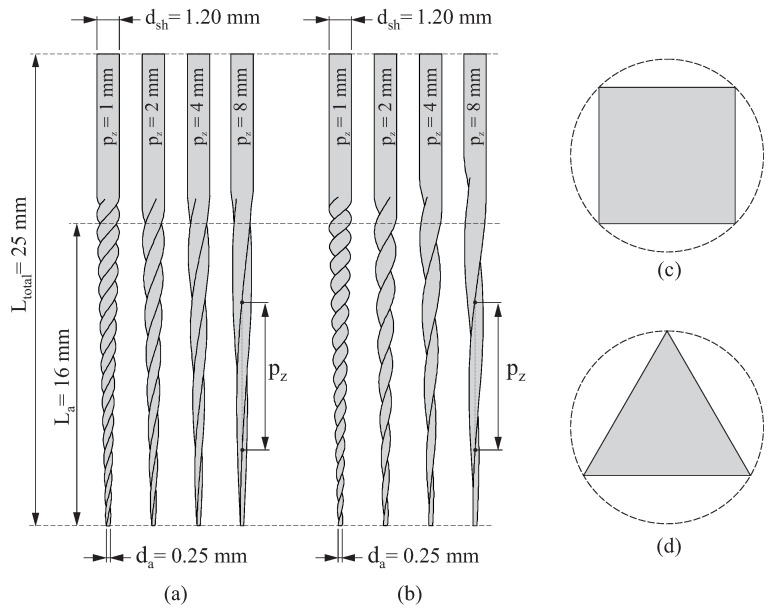
Geometries of the analyzed endodontic files: endodontic files with square cross-section (**a**); endodontic files with triangular cross-section (**b**); normalized square cross-section (**c**); and normalized triangular cross-section (**d**).

**Figure 2 jcm-11-02642-f002:**
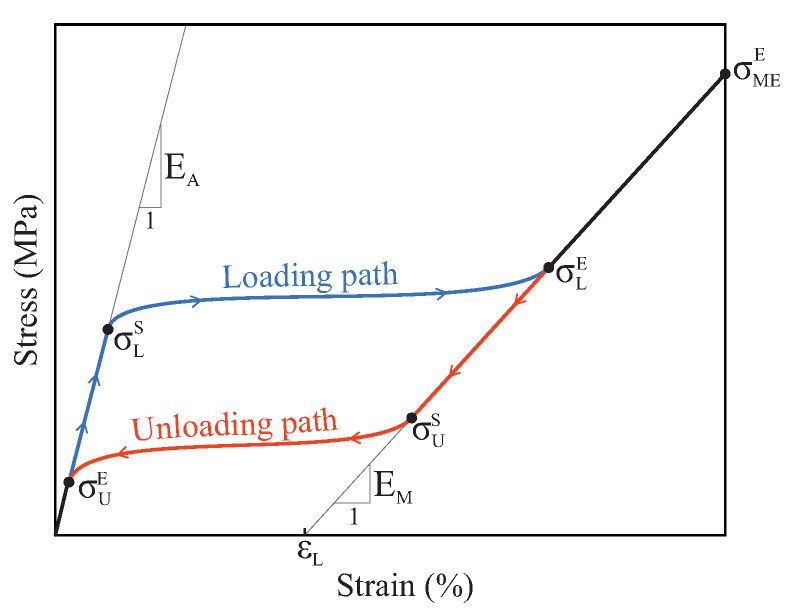
Sample stress–strain curve for NiTi material.

**Figure 3 jcm-11-02642-f003:**
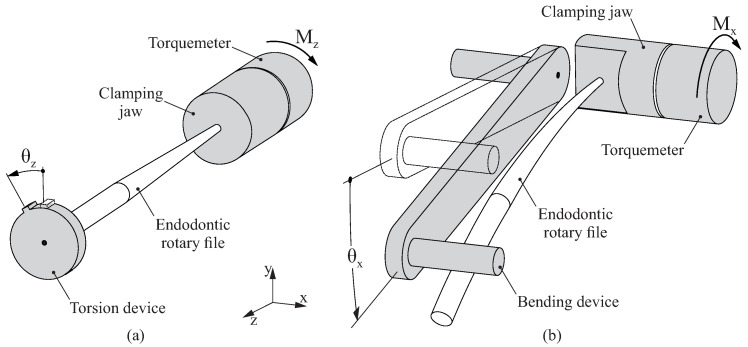
Devices used for torsion (**a**) and bending (**b**) analyses.

**Figure 4 jcm-11-02642-f004:**
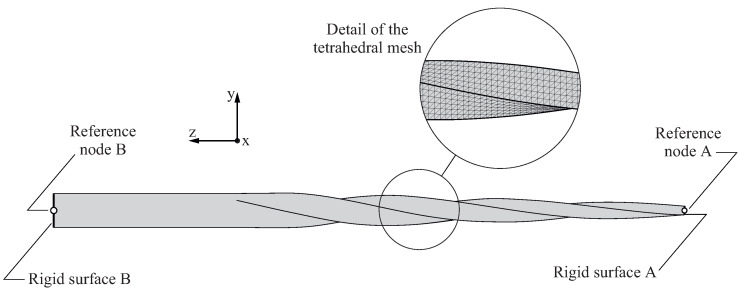
Definition of the finite element model.

**Figure 5 jcm-11-02642-f005:**
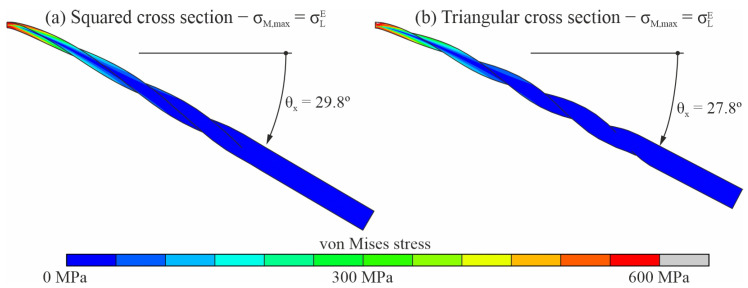
The von Mises stress plots for the bending analysis of endodontic files with $p_{z}=4\;\mathrm{mm}$ and $\varphi =0^{\circ }$.

**Figure 6 jcm-11-02642-f006:**
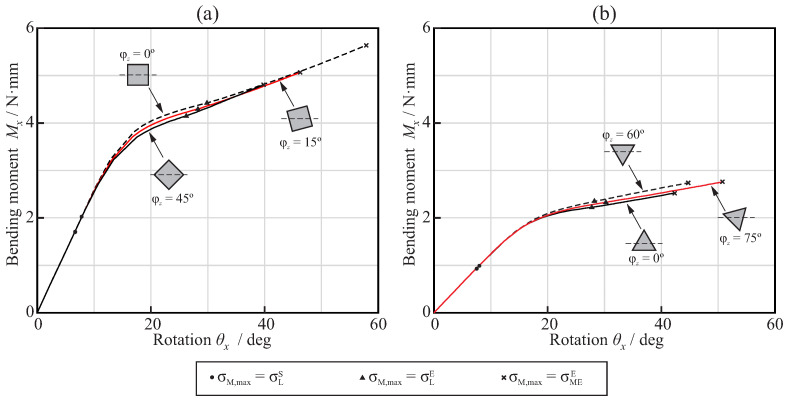
Bending moment–rotation relationships for the bending analysis of endodontic files with $p_{z}=4\;\mathrm{mm}$: squared cross-section (**a**) and triangular cross-section (**b**).

**Figure 7 jcm-11-02642-f007:**
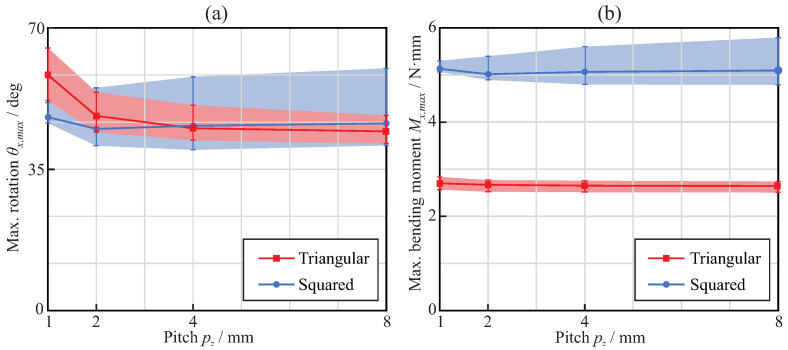
Bending analysis: effect of the pitch on the maximum rotation (**a**) and maximum applied torque (**b**) when the end of the martensitic elastic regime is reached.

**Figure 8 jcm-11-02642-f008:**
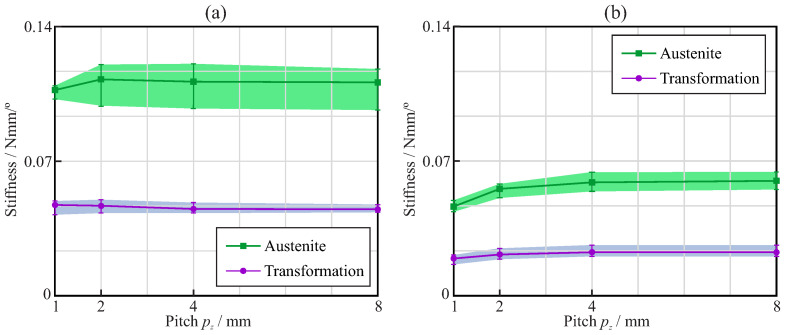
Bending analysis: bending stiffness of the endodontic rotary files with (**a**) square and (**b**) triangular cross-section.

**Figure 9 jcm-11-02642-f009:**
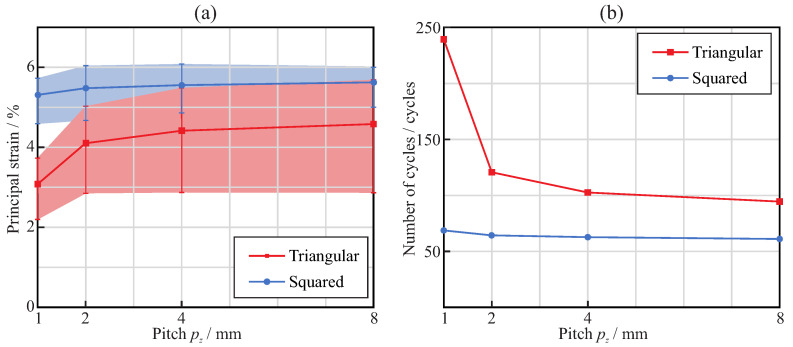
Bending analysis: effect of the pitch on the maximum principal strain (**a**) and the expected number of cycles (**b**) when the rotated angle is $\theta_{x}=20^{\circ }$.

**Figure 10 jcm-11-02642-f010:**
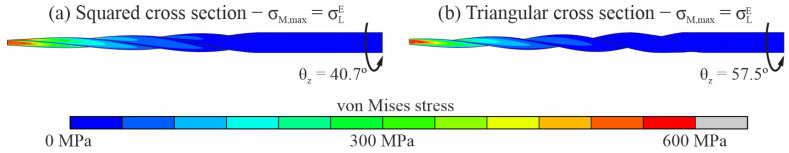
The von Mises stress plots for the torsional analysis of endodontic files with $p_{z}=4\;\mathrm{mm}$.

**Figure 11 jcm-11-02642-f011:**
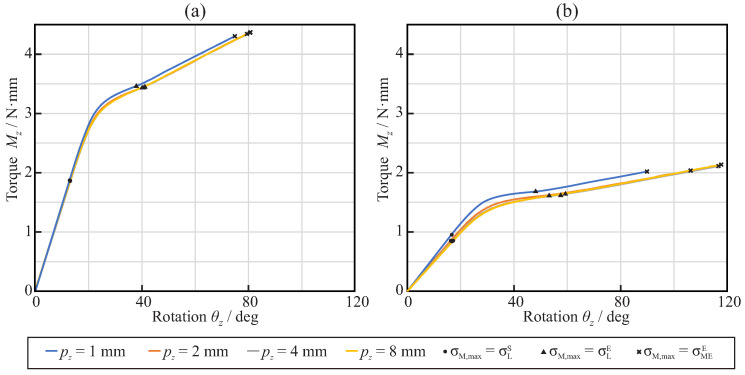
Torque–rotation relationships for the torsional analysis of endodontic files with different axial pitch: squared cross-section (**a**) and triangular cross-section (**b**).

**Figure 12 jcm-11-02642-f012:**
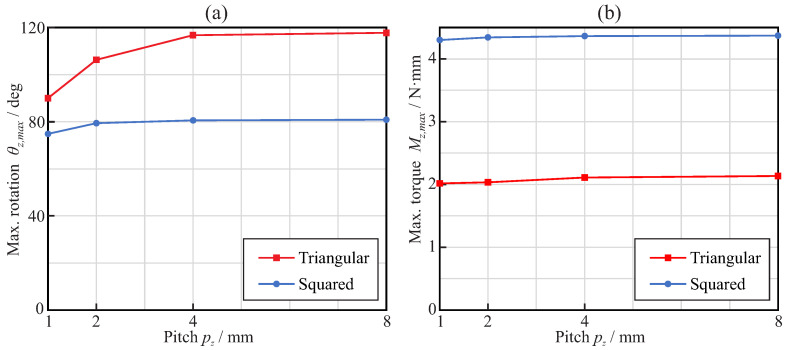
Torsional analysis: effect of the pitch on the applied torque (**a**) and rotation (**b**) when the end of the martensitic elastic regime is reached.

**Figure 13 jcm-11-02642-f013:**
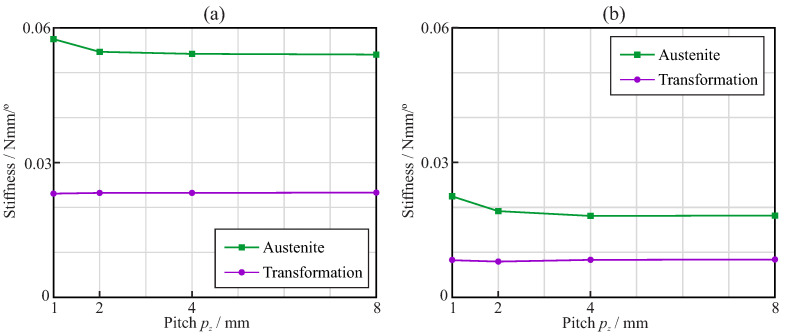
Torsional analysis: torsion stiffness of the endodontic rotary files with (**a**) square and (**b**) triangular cross-section.

**Table 1 jcm-11-02642-t001:** Material properties to characterize the super-elastic behavior of NiTi alloy. Reprinted/adapted with permission from Ref. [10]. 2014, Elsevier.

Parameter	Variable	Magnitude
Young’s modulus of austenite	$E_{A}$	42,530 MPa
Austenite Poisson’s ratio	$\nu_{A}$	0.33
Young’s modulus of martensite	$E_{M}$	12,828 MPa
Martensite Poisson’s ratio	$\nu_{M}$	0.33
Uni-axial transformation strain	$\varepsilon_{L}$	6%
Slope of the stress–temperature curve for loading	${\left(\delta \sigma /\delta T\right)}_{L}$	6.7
Start of transformation loading	$\sigma_{L}^{S}$	492MPa
End of transformation loading	$\sigma_{L}^{E}$	630MPa
Reference temperature	$T_{0}$	22 ${}^{\circ }$C
Slope of the stress–temperature curve for unloading	${\left(\delta \sigma /\delta T\right)}_{U}$	6.7
Start of transformation unloading	$\sigma_{U}^{S}$	192MPa
End of transformation unloading	$\sigma_{U}^{E}$	97MPa
End of martensitic elastic regime	$\sigma_{ME}^{E}$	1200MPa

**Table 2 jcm-11-02642-t002:** Material properties used to characterize the fatigue behavior of NiTi alloy [28,40].

Parameter	Variable	Magnitude
Fatigue ductility coefficient	$\varepsilon_{F}^{{}^{\prime }}$	0.68
Fatigue strength coefficient	$\sigma_{F}^{{}^{\prime }}$	705MPa
Fatigue ductility exponent	*c*	−0.6
Fatigue strength exponent	*b*	−0.06
Modulus of elasticity	*E*	42.5GPa

## Data Availability

Not applicable.

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
