# Peer review of "Influence of Cross-Section and Pitch on the Mechanical Response of NiTi Endodontic Files under Bending and Torsional Conditions—A Finite Element Analysis"

_jcm, 2022, doi:10.3390/jcm11092642_

Round 1

Reviewer 1 Report

The manuscript describes the influence of cross-section and pitch on the mechanical response of NiTi endodontic files under bending and torsional conditions with a finite element method. 

This topic has an important clinical value and needs to be deepened to increase clinical success and produce the ideal endodontic instrument.

The described protocol is interesting and well-conducted. Although, some corrections are required.

The authors did not specify the kind of motion applied (continuous rotation, reciprocation and adaptive motion). This is a really important aspect that need to be analyzed because of its influence on clinical success. It has been described that the type of motion may influence for example the debris extrusion or the strength of the endodontic file. 

Furthermore, many endodontic files have been proposed with a variable pitch along the file length to optimize the mechanical properties of the instrument. The authors only analyzed files with a fixed pitch and It would be interesting to study a variable one. 

Criticisms expressed above represent limitations to the study, which is pretty theoretical and could not completely fit the clinical circumstances.

Author Response

Dear Reviewer 1,

’m pleased to resubmit the manuscript of the work entitled, "Influence of Cross-Section and Pitch on the Mechanical Response of NiTi Endodontic Files under Bending and Torsional Conditions. A Finite Element Analysis".

Question Q1.0: The manuscript describes the influence of cross-section and pitch on the mechanical response of NiTi endodontic files under bending and torsional conditions with a finite element method.
This topic has an important clinical value and needs to be deepened to increase clinical success and produce the ideal endodontic instrument. The described protocol is interesting and well-conducted. Although, some corrections are required.

Response R1.0: We thank the reviewer for this comment.

Question Q1.1: The authors did not specify the kind of motion applied (continuous rotation, reciprocation and adaptive motion). This is a really important aspect that need to be analyzed because of its influence on clinical success. It has been described that the type of motion may influence for example the debris extrusion or the strength of the endodontic file.

Response R1.1: The reviewer is right: the kind of motion applied to the file can affect its strength, but only in those failure modes related to cyclic loading conditions (i.e. fatigue failure). It has been described by many authors that NiTi components are subjected to low cycle fatigue and, in this type of fatigue phenomena, the number of cycles that can withstand the endodontic file before its failure is related to the strain range through the Coffin-Manson relation. In our work the finite element analysis is used to calculate the strain range, which is obtained by observing the strain history at each point of the file during a work cycle. In this work, it is considered that the file is subjected to a continuous rotation, in which a work cycle comprises a complete revolution of the file. Under these circumstances, it has been proven in our previous works that the file undergoes a purely reversed fatigue phenomena, in which the strain history at each point of the file is similar to a sinusoidal wave
that varies between two peak values that define the strain range (being zero the average stress and identical peak values in tension and compression). This is explained in Appendix A.2.
When other file motions are considered (reciprocating or adaptive), the strain history at each point of the file cannot be represented by a sinusoidal wave and the fatigue phenomena is no longer purely reversed (average stress is no longer zero nor the peak values in tension and compression are identical). In these cases, some criterion has to be adopted to evaluate the fatigue life of the files. The study of the fatigue life of the files under these motion types is considered for further investigations, and it constitutes a limitation of this work.

In the original manuscript, it was specified that bending fatigue life was calculated assuming that the files were subjected to purely reversed cyclic loading conditions (pure reversed fatigue phenomena). However, and in order to clarify this issue, in the revised version of the manuscript the considered kind of motion (continuous motion) has been specified in those sections where the fatigue phenomena is dealt with (please, see Appendix A.2), and this issue has been identified as a limitation of this study (please, see Discussion) and included as a future work (please, see Conclusions).

Question Q1.2: Furthermore, many endodontic files have been proposed with a variable pitch along the file length to optimize the mechanical properties of the instrument. The authors only analyzed files with a fixed pitch and It would be interesting to study a variable one.

Response R1.2: As the reviewer suggests, the utilisation of a variable pitch along the length of the file can be used to optimise its mechanical response under different kinds of loading and boundary conditions. In fact and in the long term, this is one of the objectives of our ongoing investigation: to develop computational tools that allow us to improve the design of the endodontic files and to keep pursuing the idea of the “ideal endodontic instrument”. However, at this point of the investigation we were concerned with studying the effect that the pitch and the cross section have over the mechanical response of the files. For this reason, we have tried to isolate each one of these parameters so we can better understand their independent effects over the mechanical response of the endodontic instrument.

In our previous work, we developed a computational tool that allows us to obtain finite element models of the files from its design parameters. This tool allows us also to consider variable pitches, so we plan to conduct these types of studies in further works. This intention has been stated in the conclusions section of the manuscript.

Question Q1.3: Criticisms expressed above represent limitations to the study, which is pretty theoretical and could not completely fit the clinical circumstances.

Response R1.3: We thank the reviewer for his comments and suggestions. We consider that stating these limitations in the manuscript will help the potential reader to understand the scope of this work, and we take note to develop these suggestions in further investigations. Moreover, in the reviewed version of the manuscript we have tried to emphasise the clinical implications of the results of this study.

We take this opportunity to thank the recommendations and suggestions made by the reviewers to improve the document.

Yours sincerely,

Reviewer 2 Report

Thanks for the interesting research.

In practical dentistry, great importance is attached to the technical characteristics of rotary instruments, information about which can improve the quality of treatment and safety.

The design of the study is well planned, but I think it's fair to ask about the number of instruments you've tested for different types of workloads. It is not clear from the description whether these were single measurements or a series

It would also be appropriate to give clinical recommendations on the use of this type of instrument, taking into account the strength characteristics.

Author Response

Dear Reviewer 2,

I’m pleased to resubmit the manuscript of the work entitled, "Influence of Cross-Section and Pitch on the Mechanical Response of NiTi Endodontic Files under Bending and Torsional Conditions. A Finite Element Analysis".

Question Q2.0: Thanks for the interesting research. In practical dentistry, great importance is attached to the technical characteristics of rotary instruments, information about which can improve the quality of treatment and safety.

Response R2.0: We thank the reviewer for this comment.

Question Q2.1: The design of the study is well planned, but I think it's fair toask about the number of instruments you've tested for different types of workloads. It is not clear from the description whether these were single measurements or a series

Response R2.1: We are sorry about this misunderstanding. As it is said in the title of the paper and the first sentence of the abstract, this is a finite element study, so no physical instrument was tested experimentally. In this study we analysed the effect of cross-section and pitch of the files through simulations with an accurate finite element model. As it is explained in the introduction and supported by several references, this numerical technique has proven to be reliable for analysing the actual mechanical response of rotary endodontic files, providing some benefits over experimental tests for understanding the effect of independent parameters, because only these specific parameters are changed in the model and because the stress and strain distribution can be thoroughly analysed. We have added a sentence at the beginning of the ‘Material and Methods’ section to state clearly that this is a finite element study and to avoid further misunderstandings in this regard.

Question Q2.2: It would also be appropriate to give clinical recommendations on the use of this type of instrument, taking into account the strength characteristics.

Response R2.2: This clinical recommendation is discussed in the penultimate paragraph of the Discussion (lines 380-388). Our results suggest that the use of instruments with triangular shaped sections and small pitch could be beneficial. Despite the explanation for this being stated in the abstract of the manuscript, we have rewritten this part in the new version, to highlight the clinical implications of our study. Also we have included these ideas in the Conclusions section.

We take this opportunity to thank the recommendations and suggestions made by the reviewers to improve the document.

Yours sincerely,

Reviewer 3 Report

Dear Roda-Casanova et al.,

The manuscript “Influence of Cross-Section and Pitch on the Mechanical Response of NiTi Endodontic Files under Bending and Torsional Conditions. A Finite Element Analysis” (jcm-1709975) by Roda-Casanova et al. studied this article the effect of the cross section and the pitch over the mechanical response of NiTi endodontic files. The topic is interesting, but I think this article should undergo proper changes in major revision. Some of my specific comments are below:

  1. In the abstract section (line 1-17), the authors should add quantitative results rather than only qualitative results.
  2. Describe the novelty of the article made by the author? From the results of my evaluation, it seems that many similar published works adequately explain what you have raised in the current manuscript related to finite element studies of endodontic files under bending and/or torsion loads. If there are something others really new in this manuscript, please highlight it more clearly in the introduction section (line 18-121).
  3. The state of the art and the significance of the current study are not clearly present, the authors should highlight it more advanced in the introduction section (line 18-121).
  4. In the introduction section (line 18-121), the authors should explain the previous research conducted and its shortcomings. It will uphold the research gap that you filled with your research novelty. I recommend the authors elaborate on their introduction section. Do not forget to attention carefully my previous comments on numbers 2 and 3.
  5. In introduction section on paragraph 2 (line 27-37) and paragraph 9 (line 107-121), please arranged it intop paragraph, not point-by-poin as current form to make it more concise.
  6. Why the present manuscript condicted bending and torsional loads to evaluate endodontic files? It should be explain more clearly in instroduction (line 18-121) since the current form not in depth.
  7. Since this manuscript conducted finite element study of metallic materials, I would encourage and advise the authors to adopt some of the specific additional references related to finite element study of metal-on-metal (line 107-121).
  8. Finite element model is missing in the materials and methods section (line 122-197) that should be presented.
  9. Meshing strategy and element type should be explained in the materials and methods section (line 122-197).
  10. Mesh sensitivity study is crucial to ensure the number of element selected in current finite elemen model. It must be explained in the present manuscript in the materials and methods section (line 122-197).
  11. Since the present manuscript perform finite elemement study, result verivication with experimental and/or analytical is crucial to make sure the current computational model is valid. And where is the validation step? Is must be presented in the beginning of results section (line 198)
  12. In the last paragraph before conclusion section (after line 379), the authors should add of one paragraph about the limitations of the presented study.
  13. The conclusion (line 273-287) of the present manuscript is not solid. Further elaboration is needed.
  14. Further research needs to be explained in the conclusion section (line 380-400).
  15. In the whole of the manuscript, the authors sometimes made a paragraph only consisting of one or two sentences that made the explanation not clearly understood. The authors need to extend their explanation to become a more comprehensive paragraph. In one paragraph, it is recommended to consist of at least 3 sentences with 1 sentence as the main sentence and the other sentences as supporting sentences.
  16. I see some errors on English in some areas of the present manuscript. To improve the quality of English used in this manuscript and make sure English language, grammar, punctuation, spelling, and overall style are correct, further proofreading is needed. As an alternative, the authors can use the MDPI English proofreading service for this issue.
  17. Please make sure the authors have used the Journal of Clinical Medicine, MDPI format correctly. The authors can download published manuscripts by Journal of Clinical Medicine, MDPI, and compare them with the present author's manuscript to ensure typesetting is appropriate. For example:
    • Journal of Clinical Medicine's logo is missing
    • Figure typesetting is not proper
    • Funding, Institutional Review Board Statement, Informed Consent Statement, Informed Consent Statement, and acknowledgment information is missing that should be put after conclusion section and before references section
    • And other

I am pleased to have been able to review the author's present manuscript. Hopefully, the author can revise the current manuscript as well as possible so that it becomes even better. Good luck for the author's work and effort.

Best regards,

The Reviewer

Author Response

Dear Reviewer 3,

I’m pleased to resubmit the manuscript of the work entitled, "Influence of Cross-Section and Pitch on the Mechanical Response of NiTi Endodontic Files under Bending and Torsional Conditions. A Finite Element Analysis".

Question Q3.0: The manuscript “Influence of Cross-Section and Pitch on the Mechanical Response of NiTi Endodontic Files under Bending and Torsional Conditions. A Finite Element Analysis” (jcm-1709975) by Roda-Casanova et al. studied this article the effect of the cross section and the pitch over the mechanical response of NiTi endodontic files. The topic is interesting, but I think this article should undergo proper changes in major revision. Some of my specific comments are below:

Response R3.0: We sincerely thank the detailed revision made by the reviewer and the comments introduced to improve the paper. Below here there is a response to each one of the specific comments.

Question Q3.1: In the abstract section (line 1-17), the authors should add quantitative results rather than only qualitative results.

Response R3.1: We have modified the Abstract in order to add quantitative results, as suggested by the reviewer.

Question Q3.2: Describe the novelty of the article made by the author? From the results of my evaluation, it seems that many similar published works adequately explain what you have raised in the current manuscript related to finite element studies of endodontic files under bending and/or torsion loads. If there are something others really new in this manuscript, please highlight it more clearly in the introduction section (line 18-121).

Response R3.2: We agree with the reviewer that several previous studies have afforded similar analysis. Because of this, we analysed them thoroughly and elaborated a table with the main information about these studies, which is included as Appendix B (cited in line 88 of the Introduction). However, all these studies have specific limitations, highlighted in the last column of that table and commented in the Introduction (lines 85-106). The model used in the present study solves the main limitations of these previous studies. We tried to explain this idea in a paragraph (lines 100-106 of the original manuscript) but we admit that the wording was not clear enough to highlight this. We have modified this paragraph to clarify this point.

Question Q3.3: The state of the art and the significance of the current study are not clearly present, the authors should highlight it more advanced in the introduction section (line 18-121).

Response R3.3: We think that the clinical relevance of the analysis undertaken is stated in the initial part of the introduction and that an extensive study of the state of the art about the effect of cross-section and pitch of the rotary instruments is provided in the Introduction and specifically through the information contained in Appendix B and lines 85-106. However, as explained in the response to your previous question, we have modified the penultimate paragraph of the Introduction in order to emphasise the significance of the present study.

Question Q3.4: In the introduction section (line 18-121), the authors should explain the previous research conducted and its shortcomings. It will uphold the research gap that you filled with your research novelty. I recommend the authors elaborate on their introduction section. Do not forget to attention carefully my previous comments on numbers 2 and 3.

Response R3.4: Previous research is already discussed in the Introduction (35 studies cited). Moreover, Appendix B includes a comprehensive review of previous studies using finite element simulation addressed to similar objectives, including a column with their main limitations. This Appendix B could serve as a good reference to be cited by other future investigations. As commented in the response to your previous questions 2 and 3, we have rewritten the penultimate paragraph of the Introduction to highlight the shortcomings in previous studies and to emphasise the significance of the present study and its clinical relevance.

Question Q3.5: In introduction section on paragraph 2 (line 27-37) and paragraph 9 (line 107-121), please arranged it intop paragraph, not point-by-poin as current form to make it more concise.

Response R3.5: We have modified these parts of the manuscript, suppressing the bullet points and connecting them to the previous paragraphs. Thanks for the suggestion.

Question Q3.6: Why the present manuscript conducted bending and torsional loads to evaluate endodontic files? It should be explained more clearly in introduction (line 18-121) since the current form is not in depth.

Response R3.6: As stated in the introduction, the fracture of the endodontic rotary files in clinical use is mainly produced by torsion overload and flexural fatigue. For this reason, bending and torsion are essential conditions to evaluate the mechanical behaviour of the endodontic instruments. This has been clarified in the second paragraph of the introduction, where the following sentence has been added: “Thus, bending and torsion are essential conditions to evaluate the mechanical behaviour of the endodontic instruments [10]”.

The assessment of the endodontic rotary files is usually conducted following the experiments indicated in the well known ISO3630 Standard, which are described in section 2.1 of the manuscript. These experiments comprise the evaluation of the strength and the stiffness of the files under bending and torsional loading conditions. Thus, in this work we have tried to reproduce the testing indications given in ISO3630 Standard using the finite element method. This is explained in the last paragraph of the introduction, where the objectives of this work are stated.

Question Q3.7: Since this manuscript conducted finite element study of metallic materials, I would encourage and advise the authors to adopt some of the specific additional references related to finite element study of metal-on-metal (line 107-121).

Response R3.7: Dear reviewer, we understand that with “finite element study of metal-on-metal” you refer to those situations in which an interaction between two bodies (contact) is considered in the finite element model. In this case there is not such interaction, and we apology for the misunderstandment. As it is typically done in these types of analyses, the clamping jaw is simulated through an encastre at the tip of the file, and the contact with the bending device is simulated through an imposed displacement at the other side of the endodontic instrument. This is explained in section 2.2. However, if the reviewer considers that any specific reference is missing in the manuscript, we will be happy to add it in the reference list.

Question Q3.8: Finite element model is missing in the materials and methods section (line 122-197) that should be presented.

Response R3.8: We think that section 2.2 of the manuscript presents all the information needed to reproduce the finite element model used in this work. The reviewer will find that in the reviewed version of the manuscript this section includes a description of the method used to generate the finite element mesh of the endodontic file (including element type and average element size), the selected material model and material properties, and the loading and boundary conditions. The appendix A also provides instructions to postprocess the results. If the reviewer considers that there is some missing information, please let us know and we will do our best to include it.

Question Q3.9: Meshing strategy and element type should be explained in the materials and methods section (line 122-197).

Response R3.9: In the authors' previous work (please, see Ref.[38]) a procedure to obtain a fully-parameterized finite element mesh of the endodontic file was proposed. Using this procedure, the finite element mesh of the endodontic file is completely defined by its geometrical parameters (diameter of the shaft, diameter of the tip of the active part, length of the active part, total length of the file and pitch of the active part) and the average element size. In the aim of brevity, this procedure is not reproduced in the present study, and only the reference to that work is provided together with the full list of parameters that allow to generate the finite element mesh (geometrical parameters in Fig.1 and average element size). In order to clarify this in the document, the following paragraph has been rewritten in the Materials and Methods section: “The geometry of the endodontic file was generated and then discretised into quadratic finite element tetrahedrons following the meshing procedure developed in authors’ previous work [38]. Using this procedure, the finite element mesh of the endodontic file is automatically built from its geometrical parameters (dsh, da, La, Ltotal and pz, given in Fig. 1) and the average element size”.

Question Q3.10: Mesh sensitivity study is crucial to ensure the number of elements selected in the current finite element model. It must be explained in the present manuscript in the materials and methods section (line 122-197).

Response R3.10: We do agree with the reviewer that conducting a mesh sensitivity study is a crucial part in the development and validation of any finite element model. In our previous work (please, see Ref.[38]) we conducted a mesh sensitivity study for similar endodontic file geometries and boundary conditions than the ones used in this one. In this mesh sensitivity study, we observed the variation of the maximum element energy error and the energy norm error as a function of the average element size in the model, and we concluded that a finite element mesh of quadratic tetrahedrons with an average size equal to 0.1 mm was a Good compromise solution between accuracy and computational cost. For these reasons, we

selected the same element type and average element size to perform this analisis. In order to clarify this in the document, the following paragraph has been added to Materials and Methods section: “To select the average element size, a mesh sensitivity study was conducted in authors’ previous work [38] for a finite element model of an endodontic file with similar geometry, element type, boundary and loading conditions as the one described in Fig. 4. In this study, the variation of the maximum element energy error and the energy norm error as a function of the average element size was observed, and it was concluded that an average element size equal to 0.1 mm provided a good compromise between accuracy and computational cost. For these reasons, this average element size was used to perform this study, resulting in a finite element model with 89295 nodes and 58749 elements”.

Question Q3.11: Since the present manuscript performs finite element study, result verification with experimental and/or analytical is crucial to make sure the current computational model is valid. And where is the validation step? Is must be presented in the beginning of results section (line 198)

Response R3.11: One of the advantages of using the finite element method to study the mechanical response of the endodontic files is that this method is able to overcome some of the limitations presented by analytical methods. As it has already been mentioned in the introduction of the manuscript, other works used analytical equations to study the deflection (using Euler-Bernoulli or Timoshenko beam theory) and the stresses (Navier law, etc.) in endodontic rotary files (i.e. those conducted by Zhang [25] and Tsao [26]). However, it is well known that these methods are only valid under situations where the strains produced in the endodontic file under load are kept within the linear range of the material, which is not the case of the files in this study. For these reasons, these analytical methods cannot be used to validate our results and we could only rely on experimental tests for validation purposes. Unfortunately, at this time we do not have the testing facilities required to produce experimental results that can be used to validate the results obtained from the finite element model. However, there are many authors (in this and in other fields) that have proven that the results obtained using the finite element method are comparable to those obtained using experimental testing devices, even considering material non-linearities as the ones included in our finite element model. Of course, some differences can be expected between the simulation results and those obtained from experimental tests, but we are convinced that those differences do not compromise the validity of our investigation. In order to clarify this issue the following paragraph has been added to the introduction section: “The ability of the finite element method to reproduce the results obtained from experimental tests of endodontic rotary files has been proven in several works [7,10,27–29], whose conclusions are summarised in a recent bibliographical review [30]”. In fact, the assessment of the mechanical response of the endodontic rotary files through finite element analyses has some advantages with respect to the experimental analyses. For example, using finite element analyses the repetitivity of the study is guaranteed while the economical costs of the analyses are reduced. Another important advantage is that it allows us to assess aspects of the mechanical behaviour of the instruments that are difficult to obtain in laboratory tests. This has also been clarified in the introduction of the revised version of the manuscript: “This review concludes that the finite element method is a reliable tool to evaluate the behaviour of NiTi rotary instruments, and has the advantage of reducing instrument development time and costs. Another important advantage of the finite element method is that it also allows us to assess aspects of the mechanical behaviour of the instruments, such as stress distribution, which are difficult to obtain in laboratory tests [10].”. Finally, we would like to emphasise that this work is focused on observing the differences in the mechanical response of the endodontic rotary files as we vary their cross section and pitch. This comparison is conducted using identical finite element models (same material properties, same mesh density and element type, same boundary conditions, etc.) with specific geometrical changes between them. For these reasons, we consider that these results are valid for comparison purposes. Notwithstanding, we have identified this issue as a limitation of this work in the discussion of the revised manuscript.

Question Q3.12: In the last paragraph before conclusion section (after line 379), the authors should add of one paragraph about the limitations of the presented study.

Response R3.12: As it has been suggested by the reviewer, a paragraph has been added before the conclusion section to indicate the main limitations of this study: “Finally, this work has certain limitations that deserve to be mentioned. This investigation has been conducted through theoretical studies by means of finite element analyses of the endodontic rotary files; no experimental tests were conducted. Regarding the investigated endodontic file geometries, all of them have uniform parameters (pitch and cross section) through their entire length, even though there are endodontic instruments in which these parameters can vary through their active length. Finally, the bending fatigue life of the endodontic instruments has been assessed considering a fully reversed fatigue phenomena corresponding to a continuous rotation motion of the file within the root canal. The study of the bending fatigue under other types of motion (reciprocating and adaptive motions) is left for further works.”.

We thank the reviewer for this suggestion, as we think that mentioning these limitations in the manuscript will provide potential readers with a better understanding of the scope of this work.

Question Q3.13: The conclusion (line 273-287) of the present manuscript is not solid. Further elaboration is needed.

Response R3.13: We do not understand this comment. Maybe it is a copy&paste problem. The conclusion is not in the lines indicated (according to the line numbers of the original manuscript). We would need a clarification before proceeding with this point.

Question Q3.14: Further research needs to be explained in the conclusion section (line 380-400).

Response R3.14: We have included potential ideas for further research in the conclusions section of the revised manuscript: “Further works on this topic could be directed to study the mechanical response of endodontic instruments with variable parameters (pitch and cross section) through their active length. The study of the bending fatigue life of the endodontic files in cases where the loading conditions does not represent a fully reversed fatigue phenomena (adaptive or reciprocating motions) also deserves attention in future investigations”. Thanks for the suggestion.

Question Q3.15: In the whole of the manuscript, the authors sometimes made a paragraph only consisting of one or two sentences that made the explanation not clearly understood. The authors need to extend their explanation to become a more comprehensive paragraph. In one paragraph, it is recommended to consist of at least 3 sentences with 1 sentence as the main sentence and the other sentences as supporting sentences.

Response R3.15: Following the recommendation of the reviewer we have reviewed the manuscript and, in order to avoid short paragraphs, we have joined all the paragraphs that presented this issue. Thanks for the suggestion.

Question Q3.16: I see some errors in English in some areas of the present manuscript. To improve the quality of English used in this manuscript and make sure English language, grammar, punctuation, spelling, and overall style are correct, further proofreading is needed. As an alternative, the authors can use the MDPI English proofreading service for this issue.

Response R3.16: As it is suggested by the reviewer, the manuscript will undergo the proofreading service provided by MDPI. Thanks for the suggestion.

Question Q3.17: Please make sure the authors have used the Journal of Clinical Medicine, MDPI format correctly. The authors can download published manuscripts by Journal of Clinical Medicine, MDPI, and compare them with the present author's manuscript to ensure typesetting is appropriate. For example:

  • Journal of Clinical Medicine's logo is missing
  • Figure typesetting is not proper
  • Funding, Institutional Review Board Statement, Informed Consent Statement, Informed Consent Statement, and acknowledgment information is missing that should be put after conclusion section and before references section
  • And other

Response R3.17: We have written the manuscript using the LaTeX template provided in the journal webpage, which was downloaded the day before we submitted our work to the editorial office (so we assume that the LaTeX template is up to date). We carefully followed the instructions provided in the LaTeX template, and we submitted the PDF obtained as compiled by Overleaf software. For these reasons, we think that the format differences with the published papers will be amended by the editorial office if the manuscript is finally accepted. In fact, we tried to change the type of LaTeX template from “submit” to “accept” and compiling errors arised due to the lack of figures that were not provided in the template. We have included the missing information (Funding, Institutional Review Board Statement, Informed Consent Statement and Informed Consent Statement) in the reviewed version of the manuscript. No acknowledgements are included in the manuscript, since this investigation was not supported by external funds. Finally, we have double checked the instructions for authors and we did not find any issue regarding the typesetting of the figures. If you would be so kind as to tell us which is the problem with the typesetting we will be pleased to amend this issue.

Question Q3.18: I am pleased to have been able to review the author's present manuscript. Hopefully, the author can revise the current manuscript as well as possible so that it becomes even better. Good luck for the author's work and effort.

Best regards,

Response R3.18: Authors would like to thank the reviewer again for providing such a detailed and profound review of the manuscript. We consider that the reviewer’s suggestions have helped us to improve the manuscript.

We take this opportunity to thank the recommendations and suggestions made by the reviewers to improve the document.

Yours sincerely,

Round 2

Reviewer 3 Report

Dear Roda-Casanova et al.,

After carefully reading the author's revised manuscript entitled "Influence of Cross-Section and Pitch on the Mechanical Response of NiTi Endodontic Files under Bending and Torsional Conditions. A Finite Element Analysis" (jcm-1709975) by Roda-Casanova et al., The authors have been made significant improvements in the revised manuscript. Also, all of the issues in my review report have been addressed precisely.

Best regards,

The Reviewer

This manuscript is a resubmission of an earlier submission. The following is a list of the peer review reports and author responses from that submission.